# Downregulation of basal myosin-II is required for cell shape changes and tissue invagination

Daniel Krueger[1,†,‡], Pietro Tardivo[1,2,†], Congtin Nguyen[1,3] & Stefano De Renzis[1,*] ORCID

## Abstract

Tissue invagination drives embryo remodeling and assembly of internal organs during animal development. While the role of acto-myosin-mediated apical constriction in initiating inward folding is well established, computational models suggest relaxation of the basal surface as an additional requirement. However, the lack of genetic mutations interfering specifically with basal relaxation has made it difficult to test its requirement during invagination so far. Here we use optogenetics to quantitatively control myosin-II levels at the basal surface of invaginating cells during *Drosophila* gastrulation. We show that while basal myosin-II is lost progressively during ventral furrow formation, optogenetics allows the maintenance of pre-invagination levels over time. Quantitative imaging demonstrates that optogenetic activation prior to tissue bending slows down cell elongation and blocks invagination. Activation after cell elongation and tissue bending has initiated inhibits cell shortening and folding of the furrow into a tube-like structure. Collectively, these data demonstrate the requirement of myosin-II polarization and basal relaxation throughout the entire invagination process.

**Keywords** basal surface; cell shape changes; myosin-II; optogenetics; tissue invagination
**Subject Categories** Cell Adhesion, Polarity & Cytoskeleton; Development & Differentiation
**The EMBO Journal (2018) 37: e100170**

See also: **BJ Thompson** (December 2018)

## Introduction

Epithelial invagination is a key morphogenetic process that contributes to the positioning of germ layers during gastrulation and shaping of organs during later stages of embryogenesis (Schock & Perrimon, 2002; Martin & Goldstein, 2014; Pearl *et al*, 2017). Over the course of the last two decades, the internalization of mesodermal cells during *Drosophila melanogaster* gastrulation, which is usually referred to as ventral furrow formation, has emerged as a powerful system to dissect the mechanisms controlling tissue invagination (Kolsch *et al*, 2007; Martin *et al*, 2009; Mason *et al*, 2013; Chanet *et al*, 2017). Both experimental and theoretical work point toward a key role of apical constriction in generating the initial forces necessary to drive invagination (Conte *et al*, 2009; He *et al*, 2014; Guglielmi *et al*, 2015). However, the causal relationship between apical constriction and the subsequent cellular and tissue processes associated with further stages of invagination remains poorly understood. Furthermore, several modes of invaginations have been identified, which seem to be independent of apical constriction. They are characterized by cell shape changes, such as apico-basal cell shortening and basal wedging (Ybot-Gonzalez & Copp, 1999; Sherrard *et al*, 2010; Kondo & Hayashi, 2015; Pearl *et al*, 2017), which are observed also during late stages of ventral furrow invagination. How these processes are dynamically integrated and regulated on the cellular and molecular level is not well understood.

Ventral furrow formation starts shortly after the onset of zygotic transcription when a monolayer of epithelial cells on the ventral side of the embryo begins to accumulate myosin-II at the apical side, while disassembling myosin-II at the basal side (Sweeton *et al*, 1991). Within a few minutes, cells constrict apically and elongate along the apical–basal axis forming a thickened placode, which is then transformed into a shallow groove as tension in the epithelium increases. Immediately after this phase, as the tissue is further internalized and folded into a tube-like shape, cells shorten along the apical–basal axis and expand at the base, acquiring a characteristic wedge form. At the same time, lateral ectodermal cells move toward the ventral midline and close off the furrow (Dawes-Hoang *et al*, 2005).

These stereotypic changes in cell morphology might reflect complex intracellular regulations, for example, of membrane and microtubule dynamics, or alternatively, passive response to apical constriction, as suggested by a recent computer simulation study (Polyakov *et al*, 2014). According to this model, which is based on region-specific elasticity of cell membranes, during the initial stages of ventral furrow invagination, apical constriction causes viscous

1 European Molecular Biology Laboratory, Heidelberg, Germany
2 IMP, Vienna, Austria
3 Northeastern University, Boston, MA, USA
*Corresponding author. Tel: +49 6221 387 8109; Fax: +49 6221 387 8166; E-mail: stefano.derenzis@embl.de
†These authors contributed equally to this work
‡Candidate for Joint PhD degree from EMBL and Heidelberg University, Faculty of Biosciences

hydrodynamic flows of the cytoplasm and movement of nuclei toward the base. Because cell volume is constant throughout the entire course of invagination, the generation of basal cytoplasmic flows causes stretching of the lateral surface and lengthening of the cells along the apico-basal axis. This first phase is concomitant with an initial bending of the tissue and formation of a shallow groove. In a second phase, the progressive loss of myosin-II from the basal surface causes cells to expand at the base and shorten along the apico-basal axis. This sequence of cell shape changes is thought to generate an elastic energy that is initially accumulated during cell lengthening and is then released during cell shortening, providing the necessary force required for completing invagination. Despite being a very attractive model because of its simplicity, so far it has not been possible to test it experimentally. Indeed, this model is based on a temporally regulated change in basal elasticity which would trigger cell shortening. The observation that myosin-II is depleted from the basal surface of invaginating cells provides support to this model (Dawes-Hoang *et al*, 2005). However, no mutation that interferes specifically with the loss of basal myosin-II has been identified, thus making it difficult to test the role of basal myosin-II downregulation during invagination. In addition, it remains unknown whether the lateral plasma membrane could display elastic-like properties on the time-scale of invagination (~10 min).

In this study, we have employed optogenetic stimulation of Rho signaling to quantitatively control myosin-II levels at the basal surface of invaginating cells during different stages of ventral furrow formation. We calibrated our system in such a way that we could reach a ~1.6-fold increase in myosin-II levels in < 5 min and maintain constant pre-invagination levels over time. This subcellular optogenetic system in combination with quantitative 4D imaging allowed us to test the consequences of inhibiting the normal reduction of basal myosin-II during invagination, both at the single-cell and at the tissue level. Our data reveal that if myosin-II levels are restored prior to the initiation of tissue bending, apico-basal cell lengthening slows down and tissue invagination is halted. If ventral cells are first allowed to complete the elongation process and form an initial groove, up-regulation of basal myosin-II inhibits cell shortening and further invagination. Collectively, our data demonstrate the role of basal myosin-II depletion during tissue invagination and support a model in which no active force other than apical constriction is required to drive the complete process of invagination.

## Results

### Quantitative modulation of myosin-II levels at the basal surface of *Drosophila* cells at the onset of gastrulation

In order to maintain constant myosin-II levels at the basal surface of ventral mesodermal cells during invagination, we employed the CRY2/CIB1 protein heterodimerization module (Kennedy *et al*, 2010; Guglielmi *et al*, 2015; Guglielmi & De Renzis, 2017) to control the plasma membrane localization of the Rho1 exchange factor RhoGEF2 (Hacker & Perrimon, 1998) using light (see scheme in

---

**Figure 1.  Optogenetic modulation of myosin-II levels at the basal surface of epithelial cells during early *Drosophila* embryogenesis.**

A    Schematic representation of the RhoGEF2-CRY2/CIBN optogenetic system employed to control myosin-II activity during early *Drosophila* embryogenesis. The photosensitive domain of CRY2 is fused to the catalytic domain of the GTP Exchange factor RhoGEF2, while CIBN is anchored at the plasma membrane. In the dark, RhoGEF2-CRY2 is present in the cytoplasm (left). Blue light illumination triggers the CRY2/CIBN interaction and causes the translocation of RhoGEF2-CRY2 to the plasma membrane, where it activates endogenous Rho1 signaling (right), and myosin-II.

B    Multiphoton microscopy ($\lambda$ = 950 nm) enables the selective illumination of the basal surface of the cells at a tissue depth > 30 μm with subcellular precision.

C    Still frames from time-lapse recordings of an embryo expressing a myosin-II probe (Sqh::GFP). Embryos were mounted vertically to image the transverse cross section using two-photon microscopy. At the onset of gastrulation, myosin-II localized to ring structures representing the leading edge of the cellularization front (lower arrow). During ventral furrow formation (torques open rectangle), myosin-II accumulated at the apical side (upper arrow) of the cells that invaginate and the basal pool was progressively depleted. Scale bar, 40 μm.

D    Quantification of basal myosin-II levels (*N* = 3 embryos) in ectodermal and mesodermal cells. In the mesoderm, basal myosin-II levels decreased 5-fold over the course of internalization, while in the ectoderm only ~3-fold. Additionally, myosin-II depletion appeared to be accelerated in the mesodermal cells as indicated by the steeper slope of the polynomial fit ($3^{rd}$ degree with $r^2_{Ectoderm}$: 0.98 and $r^2_{Mesoderm}$: 0.87). Note the logarithmic *y*-axis.

E–H   Embryos at the end of cellularization co-expressing RhoGEF2-CRY2, CIBN::pmGFP, and Sqh::mCherry were mounted with the dorsal epithelium facing the objective, and a region of interest (red dashed line) was defined to specifically illuminate the cell base using two-photon microscopy. A representative embryo (*N* = 3) showing basal myosin-II from top view (sum-of-slice projection of 6 focal planes) at the initial stage of the experiment (E) and 6 min after activation (F). *Z*-projection of the same embryo showing the cross section of the activated region (AR) (G) and non-activated region (NR) (H) with the superimposed plasma membrane signal (CIBN::GFPpm in magenta) recorded immediately after the final Sqh::mCherry acquisition (displayed in white). The contrast has been adjusted to better visualize the distribution of myosin-II along the apico-basal axis in the respective area of the tissue. Scale bars, 20 μm.

I     Quantification of myosin-II levels (*N* = 3) within the region of activation (AR, red) relative to the non-activated region (NR, green) over the course of 6 min. The *y*-axis represents the fold change of myosin-II signal intensity in the activated region versus non-activated region. A polynomial function ($2^{nd}$ degree, $r^2_{AR}$ = 0.88) was fitted to the data (straight line) indicating the level of saturation which is reached at about 1.6-fold change of myosin-II in the activated area after 6 min.

J     Analysis of the extent of signal spreading in the *z*-dimension (apico-basal) of basal myosin-II. Myosin-II signal intensity was plotted along the apico-basal axis and a Gaussian curve fitted to the basal peak. The width of the Gaussian fit was used as a parameter to estimate myosin-II signal spreading in the *z*-dimension, which was limited to 4 μm (*N* = 3). In each box plot, the central mark, the bottom, and the top edge of each box indicate the median, the $25^{th}$ percentile, and $75^{th}$ percentile, respectively. Whiskers extent to the most extreme data point.

K     Quantification of the diameter of basal actomyosin rings in the non-activated region (NR, green) and activated region (AR, red) showing increased speed (4.8-fold change) and extent of ring constrictions in the photo-activated region (*n* = > 41 cells). The ring diameter was normalized to the mean value of the first time point. The constriction speed was estimated by the slope of a fitted linear function to the data.

L     Light activation of the cell base resulted in a compaction of the tissue in the activated (AR) compared to the non-activated region (NR). Actomyosin rings were counted and normalized to the analyzed area to score ring density, which increased 1.35 times in the activated region after 5 min of photo-activation, showing that cells were still interconnected at the base and that photo-activation did not affect tissue integrity (*N* = 3 embryos; $n_{NR}$: 431, $n_{AR}$: 302). In each box plot, the central mark, the bottom, and the top edge of each box indicate the median, the $25^{th}$ percentile, and $75^{th}$ percentile, respectively. Whiskers extent to the most extreme data point.

Data information: When present, *$P \leq 0.05$, n.s. indicates no statistically significant differences according to two-sample *t*-test.

Fig 1A). This system allows for the selective stimulation of Rho signaling at the apical surface of *Drosophila* epithelial cells, resulting in the apical accumulation and activation of myosin-II in a light-dependent manner (Izquierdo *et al*, 2018). Using this method, we set up conditions to achieve an increase in myosin-II levels at the basal surface of ventral cells (Fig 1B) corresponding to the amount that is lost during invagination. Quantification of myosin-II levels revealed a half-maximal reduction of ~3.5-fold at 5 min after the onset of ventral furrow initiation (Fig 1C and D and Movie EV1). However, during that time myosin-II levels dropped also at the basal

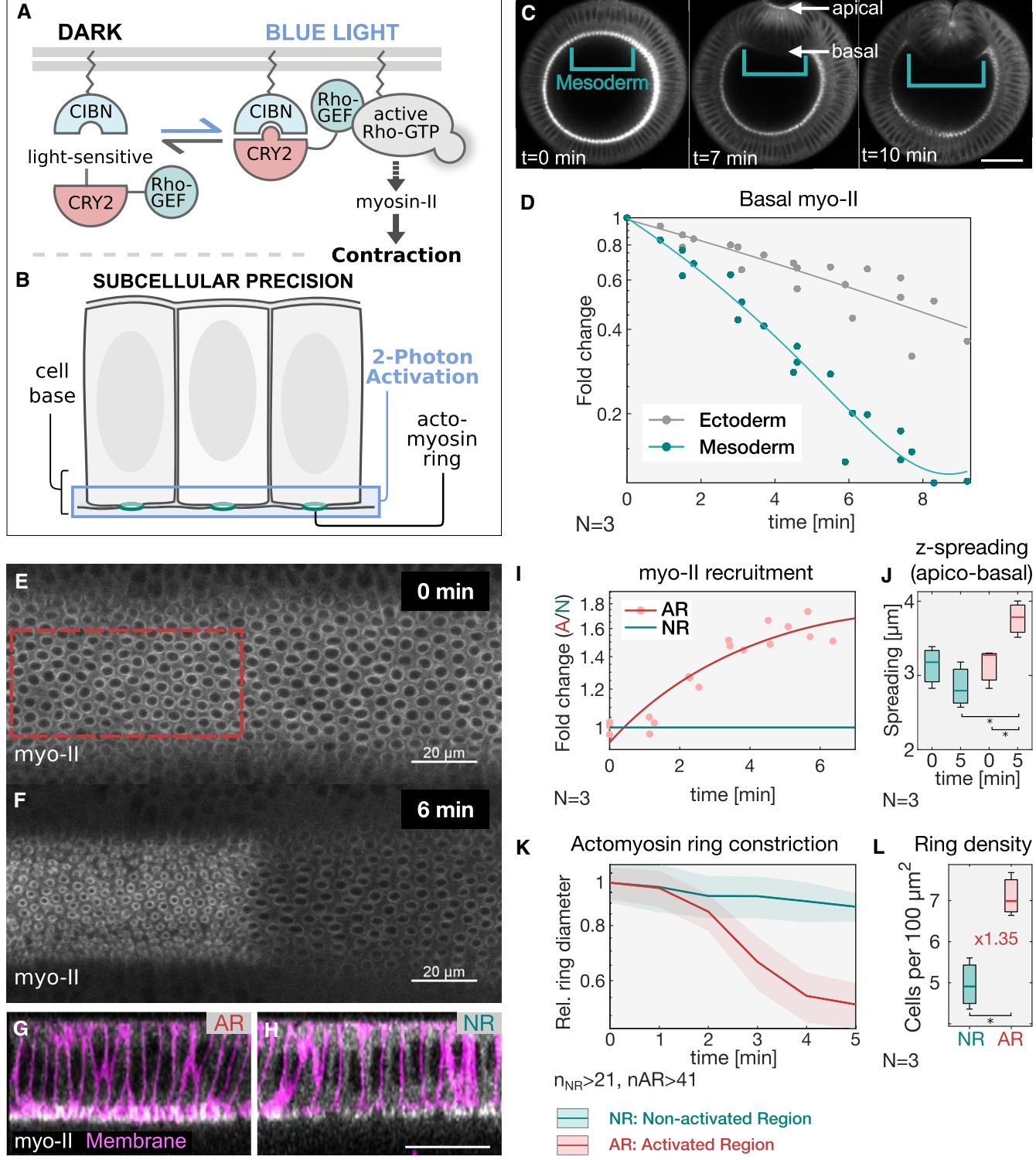

**Figure 1.**

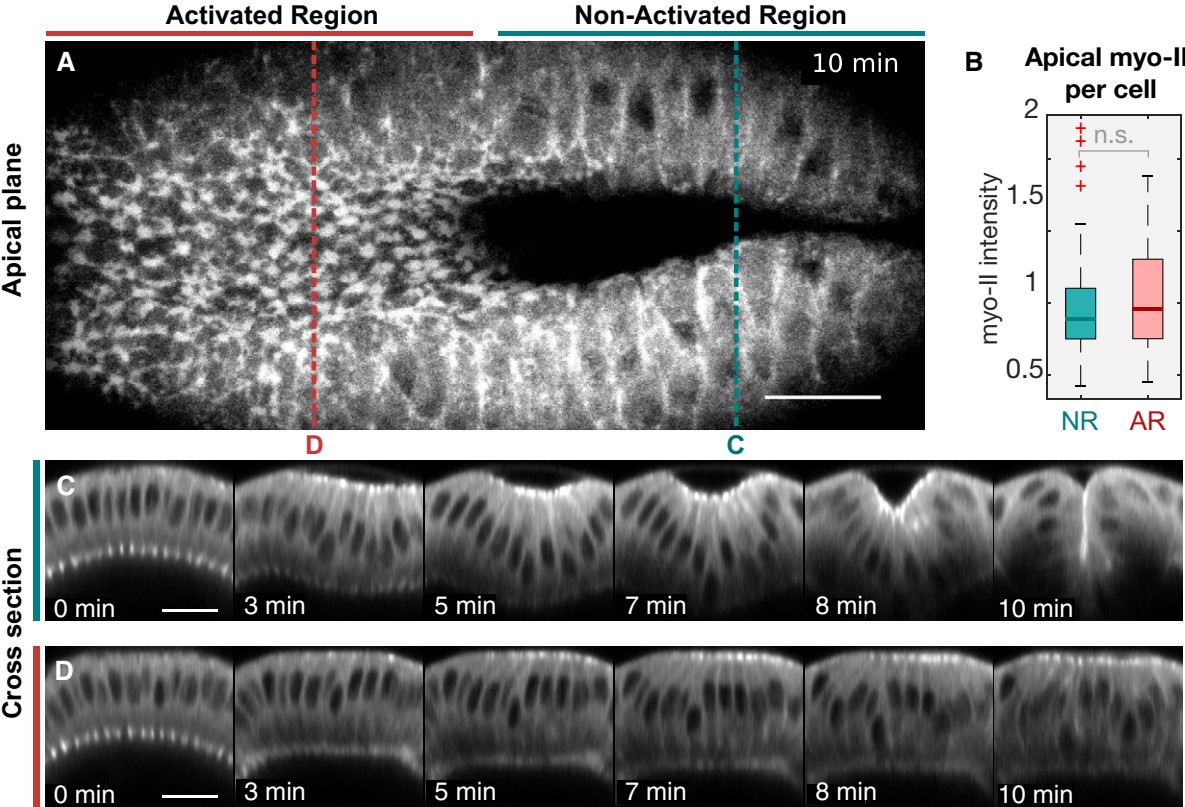

**Figure 2.    Stabilization of basal myosin-II inhibits ventral furrow formation despite normal apical myosin-II accumulation.**

Ventral furrow formation was inhibited upon photo-stabilization of myosin-II levels at the cell base. Embryos co-expressing the optogenetic module and the myosin-II probe Sqh::mCh were mounted with the ventral tissue facing the objective. A region of photo-activation (two-photon λ = 950 nm) was defined at the cell base in the anterior half of the embryo, and whole-cell myosin-II signal was recorded (confocal setup; λ = 561 nm) in an alternating fashion. Scale bars, 20 μm.

A    Top view showing apical myosin-II 10 min after the initial activation. While cells in the non-activated region have internalized and disappeared from the apical focal planes (right), cells in the activated region failed to internalize despite showing myosin-II accumulation at the apical surface (left). Red and blue dotted lines indicate the location of the cross-sectional panels displayed in (C) and (D).

B    Quantification of apical myosin-II levels demonstrates that the defects in tissue invagination were not due to impaired apical myosin-II accumulation as the average apical myosin-II level per cell did not significantly differ between the non-activated (NR) and activated region (AR). The intensity values were normalized using the median value of the combined cell population in the NR and AR ($N$ = 3 embryos; $n_{NR}$: 94, $n_{AR}$: 78). Significances (by two-sample $t$-test): n.s. indicates no statistically significant differences. In each box plot, the central mark, the bottom, and the top edge of each box indicate the median, the 25th percentile, and 75th percentile, respectively. Whiskers extend to the most extreme data point and the "+" symbol indicates an outlier.

C, D    $z$-projections (transverse cross section) of myosin-II time-lapse recordings of the non-activated (C) and activated region (D). While cells in the non-activated region internalized, resulting in a complete closure of the furrow (C), stabilization of basal myosin-II levels prevented invagination (D). Scale bars, 20 μm.

surface of lateral ectodermal cells of ~1.5-fold, resulting in a net reduction of ~2-fold in the mesoderm (Fig 1D). Therefore, in order to not exceed the levels of myosin-II present in lateral ectodermal cells, we established conditions that would allow an increase in myosin-II levels at the basal surface of invaginating cells of about 2-fold. To facilitate optogenetic calibration, we focused on the dorsal epithelium at the end of cellularization, as this tissue is not internalized and does not move at this stage. Embryos co-expressing the catalytic domain of RhoGEF2 fused to the light-sensitive protein domain CRY2 (RhoGEF2-CRY2), together with the N-terminal domain of the CRY2 binding partner CIB1, which also contained a plasma membrane anchor and GFP (CIBN::pmGFP), were photo-activated using two-photon illumination (λ = 950 nm) at the basal surface of the cells for a total volume of 5 μm. Under the established conditions, corresponding to a laser power of 13 mW and a scanning time of 60 s at 95 s intervals, myosin-II levels (visualized by the Sqh::mCherry probe; Fig 1E–H and Movie EV2) increased by ~1.6-fold over the course of 6 min (Fig 1I). During that time, myosin-II remained tightly localized at the basal surface of the cells, with a maximum spreading along the apico-basal axis ($z$-spreading) from the most basal plane of ~4 μm (Fig 1G and H and Movie EV3). This value was only slightly higher than non-activated control cells (Fig 1J), demonstrating the efficacy of this protocol in the selective up-regulate of myosin-II at the base. Consistently, RhoGEF2-CRY2 (RhoGEF2-CRY2::mCherry) displayed similar level of up-regulation and spatial distribution (Fig EV1A and B). Selective basal optogenetic activation resulted in an almost 5-fold increase in the constriction rate of the actomyosin rings present at the base of the cellularizing epithelium in 5 min (Fig 1K). To exclude the possibility that the increased actomyosin ring constriction caused a loss of basal tissue integrity, we compared the density of actomyosin rings inside and outside the photo-activation area. The result of this

measurement revealed a ~1.3-fold increase in ring density upon photo-activation (Fig 1L and Movie EV2), arguing that cells were still interconnected at the base. Thus, we conclude that the established conditions allow for the quantitative control of myosin-II levels on the time-scale of ventral furrow formation.

## Increasing myosin-II levels at the basal surface of ventral cells inhibits cell elongation and tissue invagination

Having established conditions to precisely control myosin-II up-regulation, we tested the effect of increasing the levels of myosin-II at the basal surface of ventral cells up to the levels present at the basal surface of ectodermal cells, without exceeding the levels present at the onset of invagination (corresponding to a ~2-fold up-regulation). Embryos co-expressing CIBNpm::GFP, RhoGEF-CRY2, and myosin-II (Sqh::mCherry) were photo-activated prior to the beginning of invagination in an semi-elliptical area at the basal surface of the ventral tissue (Fig EV2A and B). The remaining part of the tissue was not photo-activated and was imaged only with a 561-nm laser to visualize and quantify myosin-Il (Fig 2A and B, and Movie EV4). The result of this experiment demonstrates that while in the non-activated area internalization proceeded with normal kinetics (Fig 2C), photo-activated cells did not internalize (Fig 2D), despite displaying a normal accumulation of myosin-II at the apical surface (Fig 2A and B, and Movie EV4). Quantification of myosin-II at the basal surface of photo-activated compared to non-activated cells confirmed a ~2-fold up-regulation (Fig EV2C).

In order to quantitatively analyze the impact of increasing myosin-II levels on cell shape, photo-activation was terminated at consecutive time points and followed by an immediate z-stack collection along the entire apico-basal axis of the cells, to record the CIBNpm::GFP plasma membrane signal (Fig 3A–C). Using automated image segmentation and tracking (Khan *et al*, 2014), cell shape was reconstructed and graphically rendered with the help of a custom-made MATLAB script. The result of this experiment is illustrated in Fig 3 and demonstrates that while non-activated cells underwent the characteristic cell lengthening and shape change associated with the first phase of ventral furrow invagination (Fig 3D–F), photo-activated cells maintained a columnar shape without changing form (Fig 3G–I and J). While the apical to basal surface area ratio (A/B) in non-activated cells decreased from 1.4 to 0.25 (Fig 3K), in activated cells the average A/B ratio remained constant (Fig 3K) as cells did not undergo basal expansion (Figs 3L and EV3A). However, over time the A/B ratio became highly variable among individual photo-activated cells (Fig 3K) reflected by the presence of cells acquiring a conic shape of opposite orientation. Importantly, neither activated nor non-activated cells changed volume over time (Fig 3M), demonstrating that the principle of volume conservation was still fulfilled.

## Increasing myosin-II levels at the basal surface of ventral cells inhibits ratchet contractions

The data collected so far show that increasing basal contractility prior to the beginning of cell shape changes and invagination inhibited cell lengthening and caused cells to maintain a columnar shape. Over time, this equilibrium is broken with some cells constricting apically and expanding at the base, while some other cells acquired the opposite shape (Fig 3K). At the tissue level, this disorganized cell behavior resulted in a lack of invagination (Fig 2D) and aniso-tropic apical cell shape (ventral cells constrict preferentially along the d-v axis and acquire an elongated shape along the a-p axis of the embryo) characteristic of wild-type embryos (Martin *et al*, 2010) or in the non-activated part of the tissue (Figs 3D–F and 4A). The

---

**Figure 3.  Basal myosin-II downregulation prior to tissue invagination is required for cell lengthening.**

A–C  Embryos co-expressing the optogenetic module and Sqh::mCherry mounted with the ventral epithelium facing the objective were subjected to cycles of photo-activation of the cell base in the anterior (left) half of the embryo and confocal recording of the whole embryo. The experiment was started at the end of cellularization when initial accumulation of apical myosin-II was detected and terminated at different time points to record the plasma membrane signal (CIBN::GFPpm) in the whole embryo using two-photon illumination. Still frames visualizing the plasma membrane (CIBN::GFPpm) at a sub-apical plane at the initial time point of the experiment (A), 3 min (B), and 10 min (C) after initial photo-activation.

D–I  The CIBN::GFPpm signal was used to reconstruct cell shape in 3D as described in the Materials and Methods. Representative examples ($18 \leq n \leq 136$) of reconstructed cell shapes in the non-activated region at the initial time point (D), 3 min (E), and 10 min (F) after photo-activation show that the cells undergo apical constriction and the characteristic transition from a columnar to a conic shape. In contrast, representative examples ($83 \leq n \leq 133$) of reconstructed cell shapes in the activated region at the initial time point (G), 3 min (H), and 10 min (I) reveal that photo-activated cells did not undergo stereotypic cell shape changes and remain columnar.

J–M  Based on the reconstructed cell shapes in the non-activated (green) and activated region (red), cell length (J), apical/basal ratio (K), basal area (L), and volume (M) were quantified at different times after initial photo-activation of the cell base. (J) While cells in the non-activated region elongated from ~25 to ~40 μm, cells in the activated region showed impaired cell lengthening, both in terms of final cell length (< 40 μm) and elongation speed. ANOVA result: $F(7, 666) = 117.9$, $P = 3.8e-112$. Cohen's $D > 0.87$ between NR and AR (excluding $t = 0$ min). (K) The apical/basal (A/B) ratio is a parameter describing cell shape that corresponds to the ratio of the most apical volume divided by the most basal volume. A ratio of 1 indicates a columnar shape, a ratio < 1 indicates a conic shape, and a high ratio > 1 indicates an inverted conic shape with an apical surface that is bigger than the basal surface. In the non-activated region, the A/B ratio decreased from ~1 at the initial time point to 0.5. In the activated region, the A/B ratio spread around a value of ~ 1 indicating persisting columnar cell shape. After 10 min of photo-activation, the A/B ratio showed higher variance, spreading between ~0.5 and ~4.5 indicating the co-existence of cells acquiring different cell shapes. There were no statistically significant differences between group means within the activated cell dataset as determined by one-way ANOVA ($F(3, 384) = 1.617$, $P = 0.18$). The cartoon illustrates the cell shapes represented by different A/B ratios. ANOVA result (all datapoints): $F(7, 666) = 44.95$, $P = 4e-52$. Cohen's $D > 0.73$ between NR and AR (excluding $t = 0$ min). (L) The basal cell area of 27 μm pre-activation (pre A; $t = 0$ min) increased to 50 μm in the non-activated region (NR), while it remained constant in the activated region (AR) after 11 min. (M) Cell volume is conserved in the non-activated and activated region. Cell volume was normalized to the mean of all samples. A one-way ANOVA test was performed ($F(7, 666) = 3.34$, $P = 0.0017$) followed by a post hoc Tukey's test and Cohen's $D$ test ($d < 0.5$) both revealing no significant difference between any sample pair. (J–M) The central mark, the bottom, and the top edge of each box indicate the median, the 25th percentile, and 75th percentile, respectively. Whiskers extent to the most extreme data point, and the "+" symbol indicates an outlier. Notches indicate comparison intervals. Between $n_{NR:11\ min} = 18$ and $n_{NR:0\ min} = 136$ cells per sample with a total number of $n_{total} = 674$ cells from at least two embryos per sample and altogether 9 embryos were analyzed. In all panels, ****$P \leq 0.0001$, ***$P \leq 0.001$, and n.s. indicates no statistically significant differences according to two-sample $t$-test.

---

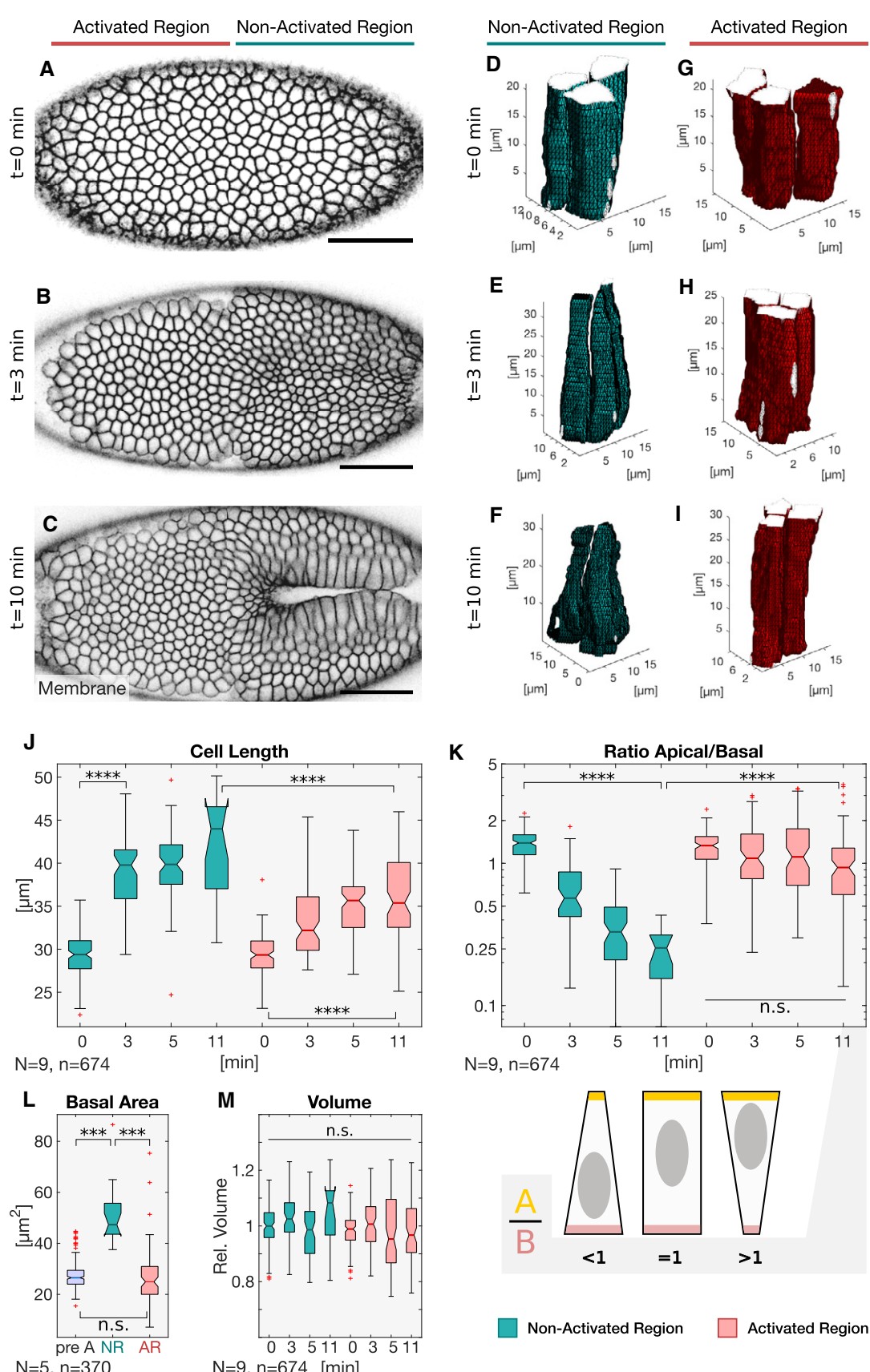

Figure 3.

**Figure 4.  Stabilizing basal myosin-II inhibits ratchet-like apical contractions.**

Embryos expressing RhoGEF2-CRY2/CIBN::GFPpm and the membrane marker GAP43::mCherry were mounted with the ventral epithelium facing the objective. To analyze apical dynamics at each activation cycle, the photo-activation time was limited to ~20 s and was followed by the acquisition of a 5-μm-sized apical image stack resulting in a total time resolution of 35 s. The basal surface of the anterior (left) half of the embryo was photo-activated.

A        Still frames of the time-lapse membrane (GAP43::mCherry) recording are shown at the initial time point (top), after 6.5 min (middle), and after 13 min (bottom) of photo-activation. Cells in the non-activated region (green) underwent normal apical constriction. In the activated region, two populations of cells were observed, one that constricted (purple) and one that did not constrict (red). Cells in the activated region that by the end of the experiment constricted to an area smaller than half of their initial area are considered as constricting cells. Cells that constricted less were considered to be defective in apical constriction and are referred to as non-constricting. Scale bar, 20 μm.

B        All cells show pulsatile behavior. Graphs illustrate the area of three representative cells in each category (non-activated region: top; activated region with constricting cells: middle; activated region with non-constricting cells: bottom) over the course of one experiment. Yellow circles indicate the position of automatically identified local maxima used for further analyses.

C, D    (C) Diagram showing the extent of ratchet-like constrictions. The mean difference in apical area between adjacent peaks (relaxed cell surface) is indicated by a red dot and reflects the extent to which cells shrink in a single pulse. Cells in the non-activated region (NR) constricted on average more than 5 μm$^2$, which is characteristic for ratchet-like constriction. Non-constricting cells in the activated region (AR, non-con.) displayed a mean difference in apical area per pulse of about 0, reflecting non-ratchet constrictions meaning that pulsing cells constricted but relaxed subsequently back to their initial size. Constricting cells in the activated region (AR, con.) show an intermediate behavior with less efficient constrictions than in the NR. Data points are significantly different as determined by ANOVA ($F_{(2, 539)}$ = 72.04, $P$ = 1.9e-28) and post hoc Tukey's test. (D) The pulsation period (T). All cells pulse with a period of about 90 s. There were no statistically significant differences between group means as determined by one-way ANOVA ($F_{(2, 540)}$ = 1.438, $P$ = 0.24). (C and D) Red dots indicate the mean values, and the error bars represent the standard deviation. More than 500 pulses ($n_{NR}$ = 119, $n_{AR, non-con}$ = 128, $n_{AR, con}$ = 295) of more than 100 cells were analyzed.

E        Diagram showing the prevalence of cells in the photo-activated region that constricted (con.) and that did not constrict (Non-con.). More than 4/5 of the cells did not constrict to half of the initial area ($N$ = 5 embryos; $n$: 407 cells; Cohen's D: $d$ = 10). In each box plot, the central mark, the bottom, and the top edge of each box indicate the median, the 25$^{th}$ percentile, and 75$^{th}$ percentile, respectively. Whiskers extent to the most extreme data point.

F, G    Graphs showing the average apical area (F) and anisotropy (G) of cells in the non-activated region (green), in the activated region that constricted (purple) and that did not constrict (red). The straight line indicates the mean value at a given time point, the semi-transparent colored area indicated the respective standard deviation. Cells in the non-activated region constricted efficiently and progressively acquired anisotropy. Constricting cells in the activated region constricted slower in an isotropic manner. Non-constricting cells in the activated region did not show net decrease in apical area and gain of anisotropy ($n_{NR}$ = 34, $n_{AR, non-con}$ = 20, $n_{AR, con}$ = 59).

Data information: In all panels, ****$P \leq$ 0.0001, and n.s. indicates no statistically significant differences according to two-sample *t*-test.

---

presence of both apically constricted and non-constricted/enlarged cells reflects either the co-existence of two distinct cell populations or of cells that oscillate between the two states (Fig EV3B). To distinguish between these two different scenarios, we followed apical surface dynamics in photo-activated embryos co-expressing CIBNpm::GFP, RhoGEF2-CRY2, and the plasma membrane marker GAP43::mCherry (Fig 4A and Movie EV5) and analyzed the pulsatile behavior of the cells (Fig 4B). The results of this analysis demonstrate that while non-activated cells displayed the characteristic cycles of constriction and stabilization of the apical surface [ratchet contractions (Martin *et al*, 2009; Xie & Martin, 2015) with a mean period of ~90 s (Fig 4C and D)]; more than 80% of the photo-activated cells (Fig 4E) underwent cycles of contractions and relaxation of the apical surface without an intervening stabilization phase and thus did not acquire a stable constricted state (Fig 4C and F). However, the remaining 20% of the cells (Fig 4E) constricted slower, revealing inhibited ratchet contractile behavior (Fig 4C and F). All photo-activated cells pulsed with the same average period of ~90 s (Fig 4D), which is comparable to the pulsatile behavior of ventral cells in wild-type embryos (Xie & Martin, 2015). However, differently from non-photo-activated cells and wild-type embryos, photo-activated cells did not acquire an anisotropic constricted shape (Fig 4G). Thus, lack of basal myosin-II downregulation interferes with the capability of cells to apically constrict, despite normal apical accumulation of myosin-II (Figs 2B and EV3A–H).

**Increasing myosin-II levels after cell lengthening and initial tissue invagination has started inhibits cell shortening and inward folding**

Next, we sought to test the effects of increasing myosin-II levels at the basal surface of ventral cells after the initial stages of

invagination, when cells had undergone apical constriction and cell lengthening already. To this end, we had to develop a strategy that allowed us to achieve precise optogenetic activation at the basal surface of the cells as the curvature of the tissue incrementally increases with the progression of invagination. One possibility was to use adaptive optics in combination with holography (Hernandez *et al*, 2016), which should allow the bending of the laser light over time in such a way that the basal surface of the cells is selectively activated at variable angles along the curvature of the tissue (Fig 5A). An alternative approach was to engineer an optogenetic anchor that is selectively localized at the basal surface of the cells and that therefore would allow for the restricted recruitment of RhoGEF2-CRY2 independently of tissue curvature and pattern of illumination (Fig 5A). We opted for this latter strategy, as it did not require changing the photo-activation protocol that was used thus far. To identify a suitable optogenetic anchor for basal recruitment, we generated a battery of fusion proteins consisting of CIBN linked to different proteins or protein domains that have been previously shown to localize to the basal surface of the cells during cellularization. Out of the six anchors tested, two were based on Bottleneck (Schejter & Wieschaus, 1993; Reversi *et al*, 2014), one on Slam (Lecuit *et al*, 2002), and three on PatJ (Lemmers *et al*, 2002; Pielage *et al*, 2003; Fig 5B). Live imaging of embryos expressing each anchor individually demonstrates restricted basal localization (Fig 5C) and efficient localized recruitment of RhoGEF2-CRY2::mCherry (Fig 5D). However, only the PatJ-based anchor (Fig 5A), consisting of PatJ fused N-terminally to CIBN with the addition of a CAAX box prenylation sequence, resulted in a corresponding increase in contractility (Fig 5E). The reasons why the other anchors tested did not support contractility, despite causing efficient recruitment, are unclear at this point, but one likely possibility is that RhoGEF2-CRY2 needs to be located at a critical distance from the plasma membrane where Rho1

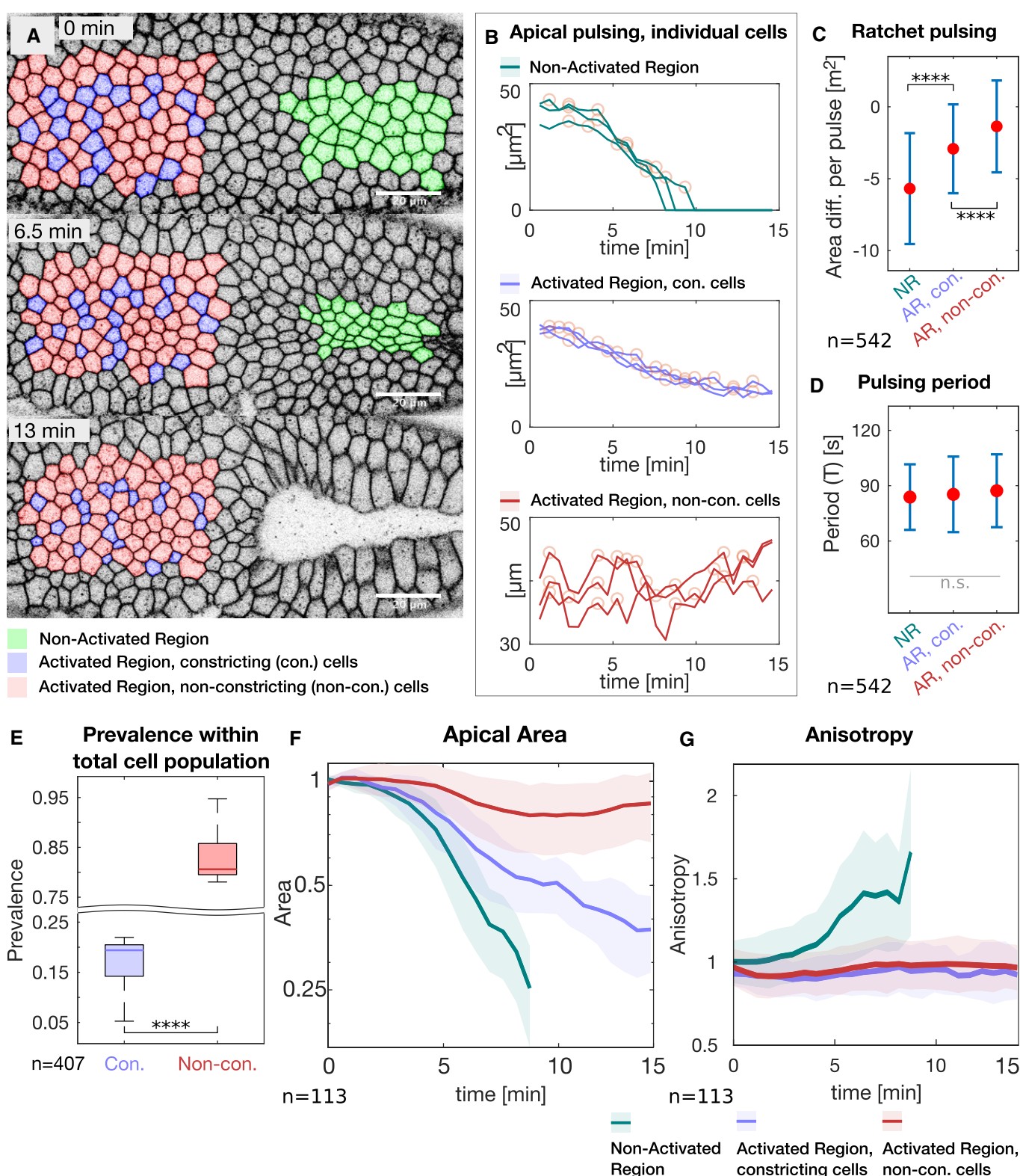

**Figure 4.**

is localized. Consistent with this hypothesis, while CIBN-PatJ without a CAAX box did not support contractility, CIBN-PatJ plus the addition of a CAAX box, which upon prenylation is directly inserted into the lipid bilayer (Powers, 1991), resulted in a twofold increase

of myosin-II levels at the basal surface (Fig EV4) and increase in contractility (Fig 6A–C). In contrast, even upon one photon illumination during early cellularization stages, when the basal surface of the forming cells is only a few microns distant from the objective and

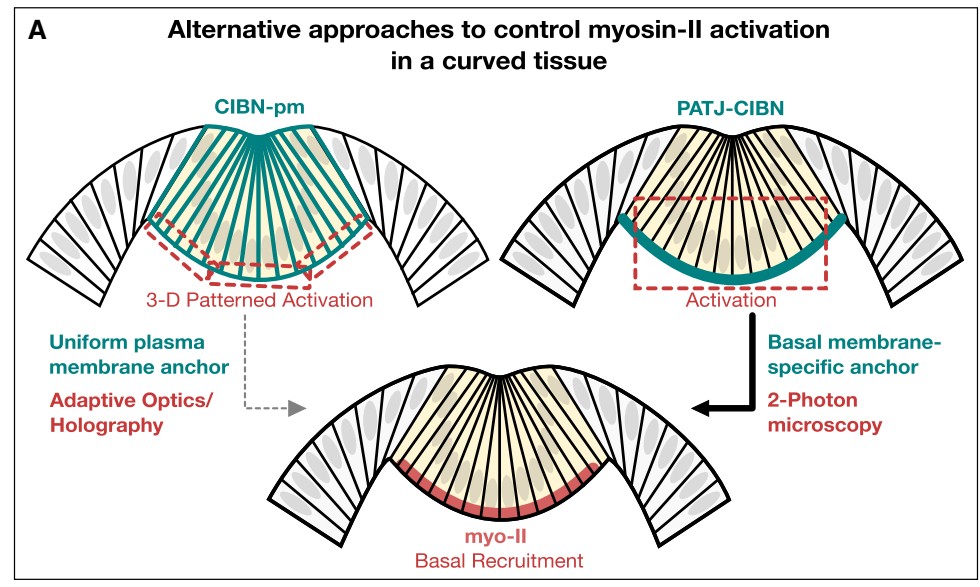

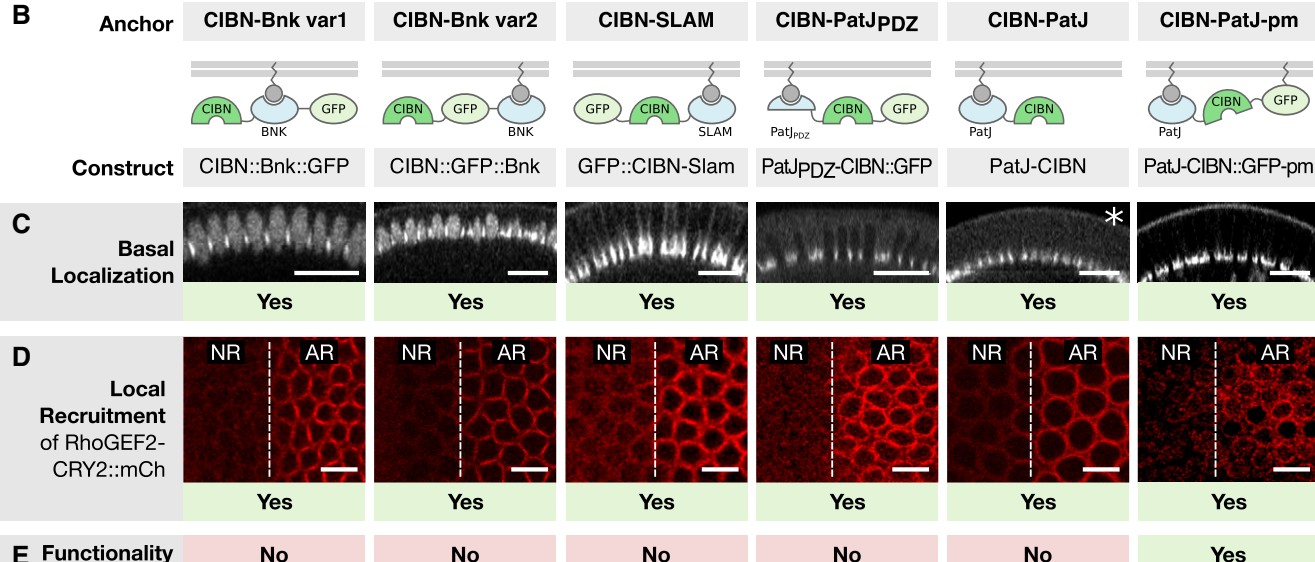

**Figure 5.  Engineering a basal-specific optogenetic anchor to enable basal-specific photo-activation in a curved tissue.**

A   Schematic illustration of two alternative approaches to control myosin-II activation in a curved tissue. In order to restrict the region of activation in a curved tissue to the basal surface using a uniformly distributed optogenetic anchor such as the CIBNpm anchor (left), the photo-activation pattern needs to account for the tissue curvature and the region of activation needs to be corrected in 3D using adaptive optics and holography. An alternative approach is to engineer an optogenetic anchor that specifically localizes to the cell base. In this scenario, a bigger focal volume can be photo-activated while still resulting in the restricted recruitment of RhoGEF2-CRY2 to the cell base.

B   Three different proteins, which have been reported to localize to the cell base during early *Drosophila* development, were expressed as CIBN fusion proteins in different configurations. Bottleneck (CIBN::Bnk::GFP, CIBN::GFP::Bnk), Slam (GFP-CIBN-Slam), and PatJ (PatJPDZ-CIBN::GFP, PatJ::CIBN, PatJ-CIBN::GFP-CAAX).

C–E   Embryos expressing either of the optogenetic anchor proteins and RhoGEF2-CRY2::mCherry were imaged during late cellularization. (C) z-projections of confocal image stacks show the localization of the respective GFP-tagged optogenetic anchor. Please note that the construct PatJ-CIBN (indicated by asterisk) lacks a fluorescent marker and its localization was thus inferred from the recruited RhoGEF2-CRY2::mCh. All anchor proteins showed predominantly basal localization during cellularization and at the onset of ventral furrow formation. Scale bars, 20 μm. (D) A region of interest (right side of the dashed line) in a 10-μm-sized stack was photo-activated using two-photon illumination for 30 s before acquiring the RhoGEF2-CRY2::mCherry signal using confocal microscopy. All anchor proteins were able to recruit RhoGEF2-CRY2::mCherry specifically to the activated region. Scale bars, 10 μm. (E) Although all basal anchors were effective at recruiting RhoGEF2-CRY2::mCherry to the cell base, only PatJ-CIBN::GFPpm resulted in effective myosin-II recruitment and actomyosin ring constriction.

thus more sensitive to photo-activation, the other basal anchors did not support myosin-II plasma membrane recruitment (Fig EV5A–L) or actomyosin ring constriction during mid-cellularization (Fig EV5M–X). Thus, lack of myosin-II recruitment provides a mechanistic explanation of why only in the presence of CIBN-PatJ-GFP-CAAX could contractility be induced.

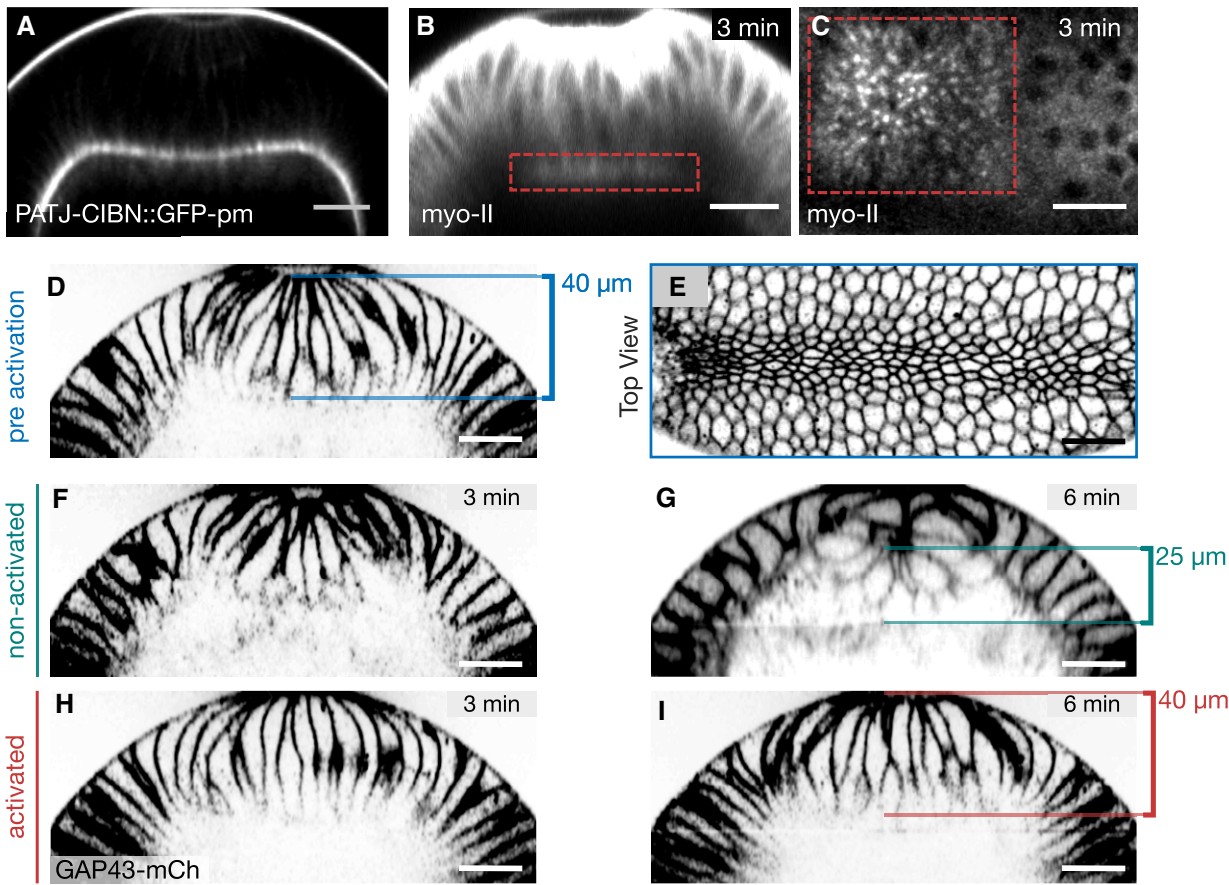

**Figure 6. Stabilization of basal myosin-II after the cell elongation phase has terminated and initial tissue invagination has started inhibits deep invagination and closure of the furrow.**

A    Localization of the basal-specific optogenetic anchor CIBN-PatJ::GFPpm during ventral furrow formation. Embryos expressing CIBN-PatJ::GFPpm were imaged using two-photon excitation during ventral furrow formation. Transverse cross section (z-projection) at a stage when cells were already apically constricted and elongated, and the ventral tissue already internalized.

B–I   Embryos co-expressing CIBN-PatJ::GFPpm, RhoGEF2-CRY2, and Sqh::mCherry (B, C) or the membrane marker GAP43::mCherry (D–I) were mounted with the ventral tissue facing the objective. After cells underwent apical constriction and cell elongation, the basal volume of the ventral cells was photo-activated using two-photon illumination and the myosin-II signal was recorded in alternation. (B) Myosin-II distribution in a cross section (z-projection) 3 min after photo-activation. Myosin-II was locally recruited to the base as indicated by the red dashed square in the activated region (red line). (C) Top view (sum of slice of 4-μm-sized image stack) showing basal myosin-II recruitment in the photo-activated region (red dashed square). (D) Cross section (z-projection) showing the plasma membrane and (E) top view showing the apical plasma membrane at the initial time point before photo-activation. At this stage, the central mesodermal tissue has started to internalize and cells were constricted at the apical surface. In an alternating fashion, the base of the cells in the anterior half of the embryo (left) was photo-activated for 100 s and the GAP43::mCherry signal in the entire field of view was recorded. The furrow further ingressed and closed completely 3 min (F) and 6 min (G) after initial activation, respectively. In the activated region, the tissue failed to internalize and to form a closed furrow both at 3 min (H) and 6 min (I) after initial photo-activation. The width of the central tissue forming the ventral furrow at the initial point of the experiment is 40 μm (D). In the non-activated region 6 min after initial activation, the width of the tissue shrank to about 25 μm (G), whereas it remained constant at 40 μm in the activated region (I).

Data information: Scale bars, 20 μm.

To study the impact of basal contractility at later stages of invagination, we generated embryos co-expressing CIBN-PatJ-GFP-CAAX hereafter referred to as CIBN-PatJ::GFPpm, RhoGEF2-CRY2, and the plasma membrane marker Gap43::mCherry. Embryos were imaged only at 561 nm to visualize cell shape and to allow ventral furrow internalization to occur. When cells started to apically constrict and invaginate, a z-stack of the Gap43::mCherry signal was recorded along the entire apico-basal axis of the cells (pre-activation control; Fig 6D and E). Next, one half of the embryo was activated at the base using two-photon illumination, while the other half, as well as the photo-activated part of the embryo, was

imaged with the 561-nm laser line to visualize the Gap43::mCherry plasma membrane signal. 6 min after the initial time point of photo-activation, the non-activated part of the embryo completed tissue invagination and the furrow folded into a tube-like shape (Fig 6F and G). In the photo-activated area of the embryo, invagination did not progress (Fig 6H and I). Quantitative analysis of cell shape (Fig 7A–C) revealed that prior to the beginning of photo-activation, cells were apically constricted and basally expanded (A/B ratio = 0.3), reaching an average length of ~45 μm (Fig 7A, D and E). While non-activated cells shorten to a length of 26 μm in 6 min accompanied by a ~2-fold increase in basal area (Fig 7B,

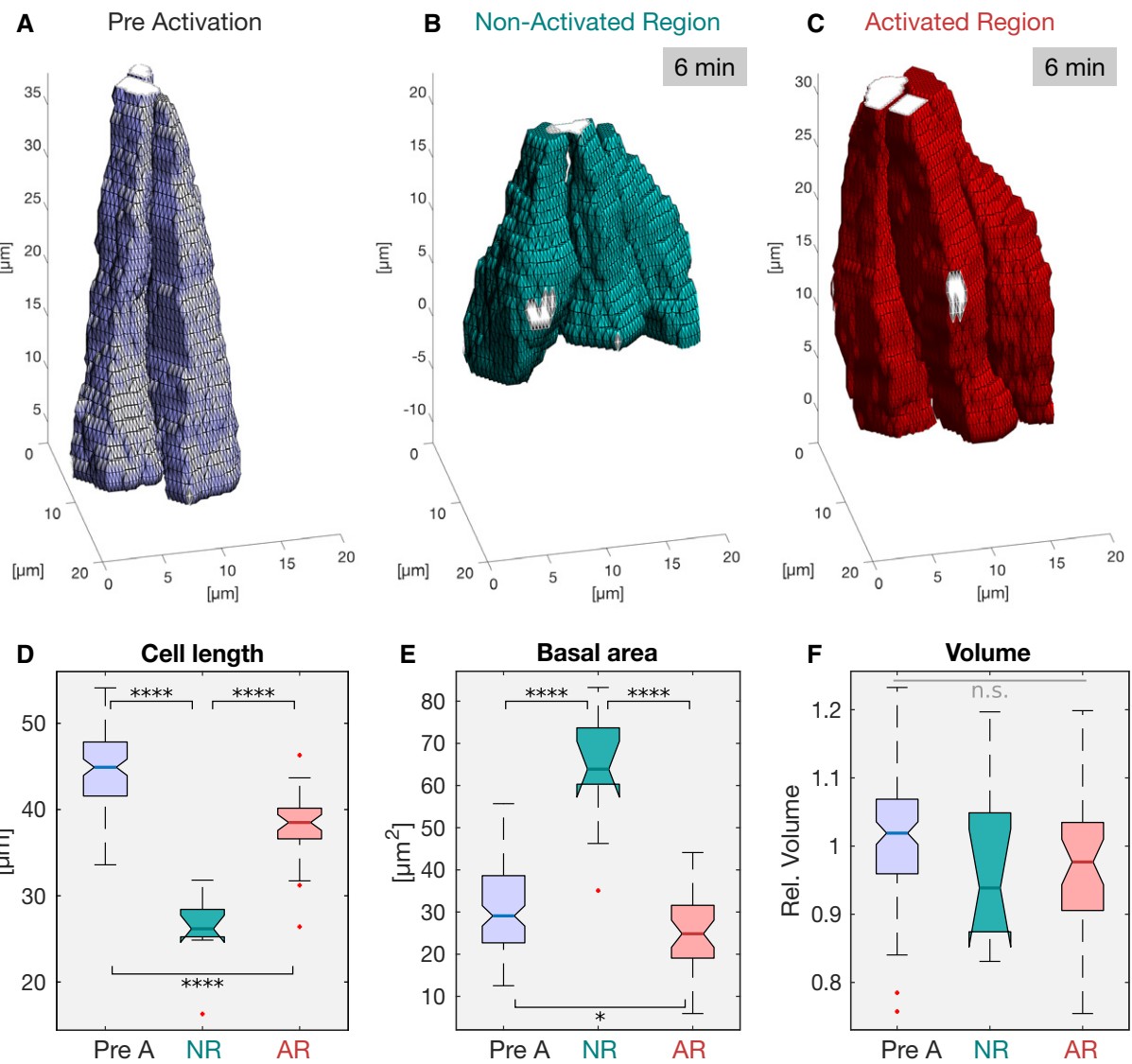

**Figure 7. Reconstruction of cell shapes reveals that myosin-II stabilization during late stages of ventral furrow formation inhibits cell shortening.**

A–C  Embryos co-expressing CIBN-PatJ::GFPpm, RhoGEF2-CRY2::mCherry, and the membrane marker GAP43::mCherry were photo-activated at the cell base after cells had already apically constricted. (A) Prior to photo-activation, cells were apically constricted and elongated. (B) In the non-activated region, within 6 min cells expanded their base and shorten. (C) In the photo-activated region, cell shortening was inhibited.

D–F  Cellular features extracted from the reconstructed cell shapes displayed in boxplots [pre A: pre-activation ($n_{pre\ A}$ = 107), initial time point; NR: non-activated region ($n_{NR}$ = 10); AR: activated region ($n_{AR}$ = 37)]. The central mark, the bottom, and the top edge of each box indicate the median, 25th percentile, and 75th percentile, respectively. Whiskers extent to the most extreme data point and the "+" symbol indicates an outlier. Notches indicate comparison intervals. (D) The mean cell length at the initial time point was 45 μm (blue). 6 min after, the non-activated cells (green) shrank to about half of the size (26 μm), whereas photo-activated cells (red) only shrank to 39 μm. ANOVA result: $F(2, 151)$ = 110.2, $P$ = 3e-30; Cohen's $D$: $d$ > 1.5 for all comparisons. (E) The basal area in the pre-activation control of 30 μm enlarged to 60 μm in the non-photo-activated cells, but remained at 30 μm in the activated region. ANOVA result: $F(2, 151)$ = 52.02, $P$ = 6.5e-18; Cohen's $D$: $d_{preA:NR}$, $d_{AR:NR}$ > 2, $d_{preA:AR}$ < 0.5. (F) The cell volume did not change between the different conditions. There were no statistically significant differences between group means as determined by one-way ANOVA ($F(2, 151)$ = 2.227, $P$ = 0.11). In all panels, ****$P$ ≤ 0.0001, and n.s. indicates no statistically significant differences according to two-sample $t$-test.

D and E), photo-activated cells shorten to only 38 μm and did not expand their basal surface (Fig 7C, D and F). In agreement with the experiments presented in Fig 3, cell volume did not significantly change over time (Fig 7F). Thus, we conclude that myosin-II downregulation is required for basal expansion and cell shortening allowing the completion of tissue invagination and closure of the ventral furrow.

# Discussion

In this study, we have employed quantitative optogenetics to precisely modulate myosin-II levels at the basal surface of invaginating cells in such a way that we could counteract its normal decrease during invagination and transform the basal surface of invaginating mesodermal cells into ectoderm-like, with respect to myosin-II

concentration. Using this approach, we have uncovered a dual role of basal myosin-II downregulation. Prior to the beginning of invagination, loss of myosin-II is required for apical constriction, cell lengthening, and invagination, whereas after invagination has started, it is needed for cell shortening and folding of the ventral furrow into a tube-like shape.

The early inhibitory effect on cell lengthening can be interpreted as secondary to the inhibition of apical constriction, which is thought to generate intracellular hydrodynamic flows resulting in basal displacement of cytoplasmic content and, given the principle of volume conservation, cell lengthening (He *et al*, 2014). Importantly, inhibition of apical constriction is not due to a lack of apical myosin-II accumulation, but rather to an incapability of cells to maintain the constricted state. Upon increasing basal myosin-II levels, apical contractions become non-ratcheted with cells constricting and expanding without a net reduction in apical area. An increase in basal contractility might, for example, interfere with the cytoplasmic flow and inhibit ratchet contractions mechanically forcing cells to pulse in a non-productive manner. However, over time some cells apically constrict and expand at the base, while some other cells acquire the opposite shape. Although we do not know what triggers these changes in cell behavior, one likely scenario is that either apical or basal contractility, initially in a few cells and possibly for stochastic reasons, prevails causing a cascade of shape changes and imbalance of forces. At the tissue level, this is manifested with the lack of the characteristic apical anisotropic constricted cell shape and invagination, thus revealing the importance of apico-basal polarization of myosin-II activity during morphogenesis of the ventral furrow. Similarly, subcellular polarization of myosin-II is observed also during cell migration (front-back; Yam *et al*, 2007) and cytokinesis (poles–cleavage furrow; Uehara *et al*, 2010), and a recent study has suggested a competitive mechanism for the accumulation of actomyosin in different parts of the cell (Lomakin *et al*, 2015). However, at the levels of optogenetic activation employed in this study (< 2-fold up-regulation of Rho signaling and myosin-II levels), we did not observe a reduction in apical myosin-II concentration. This suggests that either the threshold for a competitive mechanism is higher than 2-fold, or that disassembly of basal myosin-II during ventral furrow invagination is not simply a passive response, but is controlled by an active mechanism, possibly under the regulation of zygotic transcription.

The later effects of basal myosin-II up-regulation on cell shortening can also be interpreted in the context of volume conservation and are consistent with the cell shape changes predicted by the region elasticity theory of cell membranes (Polyakov *et al*, 2014). According to this model, the elastic energy that accumulates during cell lengthening in the lateral membrane is released during basal relaxation causing cell shortening and invagination. Given the timescale of invagination, it remains to be investigated whether this is solely an elastic actomyosin response. Interestingly, a recent study has demonstrated that stress fibers display elastic-like properties on time-scales exceeding the turnover of individual constituent components (> 60 min). This elastic-like behavior is regulated and depends on specific stress fiber proteins such as zykin (Oakes *et al*, 2017). Elastic effects on surprisingly long timescales have also been observed in cell monolayers (Sunyer *et al*, 2016). As a next step, it will be important to study the molecular architecture and functional organization of the later cortex of ventral cells as well as of cell

types involved in other modes of invagination. In principle, the effects of basal myosin-II up-regulation on cell shape could be mediated also by intracellular processes other than only mechanics. However, we think this is unlikely, given that the very same perturbation results in two opposite inhibitory effects on cell shape (lengthening and shortening), with the only difference being the timing at which optogenetic activation is initiated during invagination.

Recently, we have reconstructed epithelial invagination by optogenetic stimulation of apical constriction in dorsal cells that normally do not undergo invagination during early *Drosophila* embryogenesis (Izquierdo *et al*, 2018). In agreement with the results presented in this study, while apical constriction is sufficient to cause an invagination, it does not support folding of the internalized tissue into a tube-like shape and closure of the furrow. One possible explanation is that, despite apical accumulation of myosin-II, lack of basal downregulation becomes the limiting factor. Alternatively, it could be the specific ratio of apical/basal myosin-II that needs to be tightly controlled in order to achieve the right sequences of cell shape changes driving a complete invagination process. Several modes of invagination have been described, which seem to be independent of apical constriction and are associated with apico-basal cell shortening or basal wedging (Pearl *et al*, 2017). The optogenetic approach using a polarized anchor protein as described in this study enables us to define subcellular photo-activation patterns in curved tissues. It will provide a useful strategy to study the role of specific surface domains in morphogenesis in tissues and organs with a more complex morphology.

# Materials and Methods

### Live imaging and optogenetics

Fly crosses, cages, and embryos expressing the light-sensitive constructs were kept in the dark. Using a standard stereomicroscope equipped with a commercial red-light emitting LED lamp, stage 5 embryos were selected under halocarbon oil and prepared for live microscopy. After removing excessive oil, embryos were dechorionated with 100% sodium hypochlorite for 2 min, rinsed with distilled water, mounted onto a 35-mm glass-bottom dish (MatTek corporation), and covered with PBS. The embryos were mounted with their ventral epithelium facing the coverslip, unless described differently.

All experiments were conducted at 20°C using a Zeiss LSM 780 NLO confocal microscope (Carl Zeiss) equipped with a 561-nm diode argon laser. Two-photon excitation was achieved using a tuneable (690–1,040 nm) femtosecond (140 fs) pulsed laser (Chameleon; Coherent, Inc.) at a repetition rate of 80 MHz with $\lambda = 950$ nm in combination with a 40×/NA 1.1 water immersion objective (Carl Zeiss). Deep Amber lighting filter (Cabledelight, Ltd) was used to filter bright field illumination.

The microscope was operated using Zen Black software (Carl Zeiss) and the Pipeline Constructor Macro (Politi *et al*, 2018). Two imaging settings were used in an alternating fashion to record mCherry fluorescent signal and to photo-activate RhoGEF2-CRY2. The mCherry fluorophore was excited using 561-nm excitation laser in confocal microscope configuration. The sample was aligned, the spatial range of *z*-stacks, and the position of the cell base

was defined in the mCherry configuration. For whole-cell recordings, the standard mCherry $z$-stack size was ~55 µm with 0.94-µm interval.

An initial mCherry (pre-activation) $z$-stack was acquired, followed by cycles of photo-activation and mCherry recording with a varying number of iterations. Local photo-activation in the non-descanned/two-photon microscopy mode was achieved with 13 mW laser power and a wavelength of 950 nm. Frames were scanned bi-directionally with a total dwell time of 3.15 µs. A region of interest (ROI) was defined to locally restrict the photo-activation to one half of the embryo or a defined subset of cells. Due to the curvature of the tissue, a semi-elliptical ROI was designed to specifically restrict basal activation to the ventral cells and exclude the lateral tissue. The cell base was photo-activated within a focal volume of 5 µm for 80 frames (60 s) per activation cycle. As the cell base further ingressed during the course of the experiment, the focus was adapted by sequentially increasing the focal offset at the macro interface and by manually changing the focus during activation. The time to complete one photo-activation cycle was about 95 s. At the end of each experiment, a whole-cell two-photon $z$-stack was acquired with a standard size of 80 µm with a 1.00-µm interval and 22 mW laser power.

For assessing apical myosin-II levels, the Sqh::mCherry signal was recorded in a 15 µm $z$-stack with a laser (561 nm) power that ensures non-saturating signal acquisition. Otherwise, the previously described photo-activation protocol was maintained. The experiment was terminated after apical constriction, and an initial indentation of the non-activated tissue (3 – 5 µm) had been observed. The final Sqh::mCherry recording was immediately followed by a whole-embryo two-photon acquisition to visualize the cell membrane and reconstruct the cell outlines.

The imaging protocol was modified to analyze apical pulsatile contractions, as a higher temporal resolution for GAP43::mCherry acquisition was required. The photo-activation time was reduced to 20 s, and GAP43::mCherry was recorded in a 5-µm spanning apical $z$-stack. The final time resolution amounted to 35 s.

Sample preparation and the general microscopy/photo-activation protocol remained constant for the PatJ-CIBN::GFP/RhoGEF2-CRY2 experiments. However, the activation time, laser power, and the illuminated volume were increased to 100 s, 20 mW, and 8 µm, respectively. All laser powers were measured 1 cm from the objective.

To test the effects on myosin-II recruitment during early cellularization, when the basal actomyosin network is very close to the apical surface thus diminishing any possible light penetration constraints both for the efficiency of photo-activation and visualization of myosin-II, embryos of the appropriate genotype were photo-activated within a region of interest using a pulse of 488-nm light ensuring an efficient activation of Rho-GEF2-Cry2 at the onset of cellularization. The effects on myosin-II were assessed by recording the Sqh::mCherry signal using 561-nm excitation pre- and post-photo-activation. To test the functionality of the newly engineered basal optogenetic anchors, embryos expressing Rho-GEF2-CRY2 and the respective anchor were continuously activated using 488-nm single-photon excitation while simultaneously recording Sqh::mCherry using 561-nm light during cellularization to visualize the effects on actomyosin-II rings.

## Image and data analysis

Images were processed and analyzed using Fiji (https://fiji.sc/) and MATLAB-R2017b (MathWorks). Zeiss LSM files were imported and metadata were extracted using the lsmread function provided via GitHub by Chao-Yuan Yeh (https://github.com/joe-of-all-trades/lsmread) and the ImageJ Bio-Formats package. Image preprocessing was done using Fiji. Image segmentation and cell tracking was done using Embryo Development Geometry Explorer (EDGE) software (Gelbart *et al*, 2012) provided via GitHub (https://github.com/mgelbart/embryo-development-geometry-explorer).

The standard settings used, that were different from default, were bandpass filter thresholds of 2 and 10 µm, minimum cell size of 2 µm, and a minimum spatial and/or temporal fractional overlap of 0.4. Vectorized cells were adjusted manually if required. Cells that were correctly and completely segmented and tracked were selected and subject to further analysis. Cell area and anisotropy measurements were extracted from EDGE. Other cell features (A/B ration, cell length, cell volume) were calculated and cell shapes reconstructed based on the processed membranes and centroid information using customized MATLAB scripts. Cells displaying extreme values in the volume parameter were thresholded within a *Coefficient of Variation (C.V)* range of $C.V = 25\%$. Cells were visualized using the MATLAB function *isosurface*.

To quantify basal myosin levels, regions of interest (ROIs) were defined within the activated and the non-activated region, respectively. Each ROI was sub-divided into smaller regions to better account for the uneven morphology of the embryo, the sub-regions processed individually and finally averaged. A line intensity plot along the apico-basal cell axis results in a $z$-profile of the myosin-II signal. Fitting a Gaussian function to the $z$-profile using the MATLAB function *findpeaks*, basal myosin peaks were identified. The width of the Gaussian fit was used as a measurement for the extent of signal spreading in the $z$-dimension. Myosin-II levels result from the mean value (integrated density per area) of 5 consecutive image frames (5 µm size) centered at the basal peak position. To obtain basal myosin-II images, a window size of five frames was centered at the isolated basal myosin-II peak both, in the non-activated and activated region. All frames within the combined frameset were summed to produce a $z$-projection. The projection was normalized by the total mean value.

In order to measure constriction rates of actomyosin rings in activated versus non-activated regions, circles were fitted to basal myosin-II projections of different time points using the *imfindcircles* MATLAB function. Diameters of the fitted circles were used to approximate the actomyosin ring size and normalized to the mean value of the initial time point. A linear function was fitted to the data with the slope being a measure for the constriction speed. Compaction of the activated tissue was analyzed by manually marking the position and automated counting of actomyosin rings. The number of rings was normalized to the analyzed area resulting in the ring density value.

For quantifying apical myosin-II upon basal activation, the myosin-II signal was superimposed to the membrane signal (image stack of 15 µm). Cells were segmented and tracked and myosin-II intensity per cell area measured using EDGE software. The procedure previously described to identify myosin-II peaks was applied correspondingly to find apical myosin-II peaks. For each cell, the

myosin-II intensities of three frames (3 μm) centered at the identified apical peak were integrated. The values were normalized by the mean integrated density per cell of the combined (non-activated and activated region) population.

In order to analyze apical pulsing behavior, apical GAP43::mCherry plasma membrane signal was segmented and tracked using EDGE software and cell area values over time were extracted. Pulses in apical cell area were identified by finding local maxima using the *findpeaks* MATLAB function. The mean temporal distance between identified peaks revealed the mean pulsing period. The mean change in area between consecutive peaks describes the extent of ratchet cell constriction during a single pulse.

## Statistical analysis

Statistical analyses were performed in MATLAB (MathWorks). Two-sample Student's *t*-test was performed to determine whether two sets of data are significantly different from each other and the *P*-value was calculated. To compare multiple samples and test for significant differences, a one-way analysis of variance (ANOVA) was performed followed by a post hoc Tukey's honestly significant difference procedure. In addition, as the sample size of the present data is high, the effect size was analyzed by calculating Cohen's *d*:

$$d = \sqrt{\frac{(\overline{x_1} - \overline{x_2})^2}{s}} = \sqrt{\frac{(\mu_1 - \mu_2)^2}{s}}$$

where $\overline{x_1} - \overline{x_1}$ is the difference between the two means of the compared sample pair and *s* is the maximal standard deviation of the analyzed dataset (comprising multiple sample pairs). Cohen's *d* values of $d \leq 0.5$ are considered as low effect size, whereas $d \geq 1$ represent a large effect size and thus argue in favor of two significantly different sample populations.

## Cloning

All constructs were generated using the Gibson assembly cloning technique and standard molecular biology procedures. In brief, a pENTR/D-TOPO-derived entry vector (Thermo Fisher Scientific) was linearized by restriction enzymes or split and amplified by PCR. Using SnapGene (GSL Biotech LLC) software DNA fragments to join were designed to have overhangs at either ends and were PCR-amplified by gene-specific primers comprising 5′ extensions. The Gibson assembly reaction was performed using a custom-made enzyme mix (Gibson *et al*, 2009). The resulting pENTR constructs were sub-cloned into the pPW vector (*Drosophila* Genomics Resource Center, Bloomington, IN) using the Gateway cloning system (Thermo Fisher Scientific) following the manufacturer's instructions.

To engineer novel basal-specific optogenetic anchors, full-length PatJ$_{1-871}$ (Q9NB04-1; reference sequence: AAN11498.1), PatJ$_{PDZ}$ (L27 and first PDZ domain comprising aa 1 – 244 from reference sequence Q9NB04-1), and full-length Slam$_{1-1,173}$ (Q9VME5-1; reference sequence: AAF52374.3) were amplified from a *Drosophila melanogaster* embryo (0–12 h) cDNA library. Full-length Bnk (P40794; reference sequence: AAC46467.1) was amplified from existing plasmids (Reversi *et al*, 2014). All other fragments including CIBN (CIB1 N-terminal domain), EGFP, and the plasma membrane (pm) anchor

(CAAX box) were amplified from existing plasmids (Guglielmi *et al*, 2015) to finally assemble (3′–5′) PatJ-Linker$_{(GAGA)}$-CIBN::GFPpm, PatJ-Linker$_{(GAGA)}$-CIBN, PatJ$_{PDZ}$-Linker$_{(LAAPFT)}$-CIBN::GFP, GFP::CIBN-Slam, CIBN::GFP::Bnk, and CIBN-Bnk::GFP.

## Fly strains and genetics

All fly stocks were kept at 22°C. Transgenic lines listed below were generated by microinjection and standard procedures. Crosses were set up and maintained in the dark to generate flies of the following genotype.

To visualize myosin-II upon activation of the optogenetic module:

w[*]; P[w+, UASp>CIBN::pmGFP]/P[w+, Sqh::mCherry]; P[w+, UASp>RhoGEF2-CRY2]/P[w+, Osk>Gal4::VP16].

To visualize RhoGEF2-CRY2 local recruitment:

w[*]; P[w+, UASp>CIBN::pmGFP]/+; P[w+, UASp>RhoGEF2-CRY2::mCh]/P[w+, Osk>Gal4::VP16].

To visualize the plasma membrane upon photo-activation of the optogenetic module:

P[w+, sqhp>Gap43::mCherry]/w[*]; P[w+, UASp>CIBN::pmGFP]/+; P[w+, UASp>RhoGEF2-CRY2]/P[w+, Osk>Gal4::VP16].

Optogenetic basal-specific anchors that resulted in RhoGEF2-CRY2 plasma membrane recruitment but did not show functionality (increased contractility):

w[*]; P[w+, UASp> CIBN::GFP::Bnk]/+; P[w+, UASp>RhoGEF2-CRY2::mCh]/P[w+, Osk>Gal4::VP16].

w[*]; P[w+, UASp> CIBN-Bnk::GFP]/+; P[w+, UASp>RhoGEF2-CRY2::mCh]/P[w+, Osk>Gal4::VP16].

w[*]; P[w+, UASp> GFP::CIBN-Slam]/+; P[w+, UASp>RhoGEF2-CRY2::mCh]/P[w+, Osk>Gal4::VP16].

w[*]; P[w+, UASp> PatJ$_{PDZ}$-CIBN]/+; P[w+, UASp>RhoGEF2-CRY2::mCh]/P[w+, Osk>Gal4::VP16].

w[*]; P[w+, UASp> PatJ-CIBN]/+; P[w+, UASp>RhoGEF2-CRY2::mCh]/P[w+, Osk>Gal4::VP16].

Optogenetic basal-specific anchor that both resulted in RhoGEF2-CRY2 plasma membrane recruitment and increased contractility:

w[*]; P[w+, UASp> PatJ-CIBN::GFPpm]/+; P[w+, UASp>RhoGEF2-CRY2::mCh]/P[w+, Osk>Gal4::VP16].

To visualize myosin-II upon basal-specific recruitment of RhoGEF2-CRY2:

w[*]; P[w+, UASp> PatJ-CIBN::GFPpm]/P[w+, sqhp>Sqh::mCherry]; P[w+, UASp>RhoGEF2-CRY2]/P[w+, mat.tubulin>Gal4::VP16].

To visualize the plasma membrane:

P[w+, sqhp>Gap43::mCherry]/w[*]; P[w+, UASp> PatJ-CIBN::GFPpm]/+; P[w+, UASp>RhoGEF2-CRY2]/P[w+, mat.tubulin>Gal4::VP16].

## Fly stocks

w[*]; If/CyO; P[w+, UASp>RhoGEF2-CRY2::mCherry]/TM3, Ser. Gal4-UAS-driven catalytic DHPH domain of *Drosophila* RhoGEF2 fused to the photosensitive PHR domain of CRY2 and the fluorescent protein mCherry.

w[*]; If/CyO; P[w+, UASp>RhoGEF2-CRY2]/TM3, Ser. Gal4-UAS-driven *Drosophila* RhoGEF2 DHPH domain fused to CRY2 PHR domain.

w[*]; P[w+, UASp>CIBN::pmGFP]/Cyo; Sb/TM3, Ser. Gal4-UAS-driven CIB1 N-terminal domain (CIBN) fused to EGFP and CAAX box (membrane anchor).

P[w+, sqhp>Gap43::mCherry]/Fm7;; Sb/TM6 Tb. Spaghetti-squash promoter-driven Gap43 membrane marker fused to the fluorescent protein mCherry.

yw[*];If/CyO;P[w+, Sqh::GFP]. Myosin regulatory light chain (Spaghetti-squash) tagged with the fluorescent protein GFP.

yw[*]; P[w+, sqhp>sqh::mCherry]/CyO; Dr/TM3, Ser, Sb]. Myosin regulatory light chain (Spaghetti-squash) tagged with the fluorescent protein mCherry.

w[*]; If/CyO; P[w+, Oskp>Gal4::VP16]/TM3, Ser. Oskar promoter-driven and maternally deposited Gal4 transcription factor (Bloomington stock number 44242).

w[*]; PatJ-CIBN::GFPpm/CyO;MKRS/TM6,Tb. Gal4-UAS-driven *Drosophila* PatJ (full-length) fused to CIB1 n-terminal domain (CIBN), the fluorescent protein EGFP and the CAAX box membrane anchor.

w[*]; PatJ-CIBN/CyO;MKRS/TM6,Tb. Gal4-UAS-driven *Drosophila* PatJ (full-length) fused to CIB1 n-terminal domain (CIBN).

w[*]; PatJ$_{PDZ}$-CIBN::GFP/CyO;MKRS/TM6,Tb. Gal4-UAS-driven first two PDZ domains of *Drosophila* PatJ (PatJ$_{1-244}$) fused to CIBN and the fluorescent protein EGFP.

w[*]; GFP::CIBN-Slam/CyO;MKRS/TM6,Tb. Gal4-UAS-driven full-length *Drosophila* Slam (Slow as molasses) N-terminally fused to the fluorescent protein EGFP and CIBN.

w[*]; CIBN::GFP::Bnk/CyO;MKRS/TM6,Tb. Gal4-UAS-driven full-length *Drosophila* Bnk (Bottleneck) N-terminally fused to CIBN and the fluorescent protein EGFP.

w[*]; CIBN-Bnk::GFP/CyO;MKRS/TM6,Tb. Gal4-UAS-driven full-length *Drosophila* Bnk (Bottleneck) flanked N-terminally by CIBN and C-terminally by the fluorescent protein EGFP.

w[*]; P[w+, mat.tubulin>Gal4::VP16]; P[w+, mat.tubulin>Gal4::VP16]. Maternal tubulin promoter-driven and maternally deposited Gal4 transcription factor (Bloomington stock number 7062-7063).

Expanded View for this article is available online.

## Acknowledgements
We thank E. Izquierdo, A. Reversi, and all members of the De Renzis laboratory for helpful discussion. We thank U. Schwarz and A. Degen for critical reading of the manuscript. We thank the advanced light microscopy core facility (European Molecular Biology Laboratory (EMBL), Heidelberg) for their advice and assistance, and A. Politi for providing the *Pipeline Constructor* macro. We thank the Bloomington *Drosophila* Stock Center for providing fly stocks and the *Drosophila* Genomics Resource Center for providing cDNAs.

## Author contributions
The experiments were conceived and designed by DK, PT, and SDR Optogenetic experiments were performed by DK. PT and CN in collaboration. Reconstruction of cell shape was performed by DK. All the data were analyzed by DK, PT, CN, and SDR. DK and SDR wrote the manuscript together with input from PT and CN.

## Conflict of interest
The authors declare that they have no conflict of interest.

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
