## [Review Process File · The EMBO Journal]

Downregulation of basal myosin-II is required for cell shape changes and tissue invagination

Daniel Krueger, Pietro Tardivo, Congtin Nguyen, and Stefano De Renzis

Review timeline:	Submission date:	2nd Jul 2018
	Editorial Decision:	24th Jul 2018
	Revision received:	6th Sep 2018
	Editorial Decision:	2nd Oct 2018
	Revision received:	8th Oct 2018
	Accepted:	12th Oct 2018

Editor: Deniz Senyilmaz Tiebe

Transaction Report:

1st Editorial Decision

24th Jul 2018

Thank you for submitting your manuscript entitled 'Downregulation of basal myosin activity is required for cell shape changes and tissue invagination' to The EMBO Journal. We have now received two referee reports, which are included below.

Given the referees' positive recommendations, I would like to invite you to submit a revised version of the manuscript, addressing the comments of the reviewers. I should add that it is EMBO Journal policy to allow only a single round of revision, and acceptance of your manuscript will therefore depend on the completeness of your responses in this revised version.

REFeree REPORTS:

Referee #1:

This is an excellent manuscript. The authors use optogenetic activation of contractility to show that basal relaxation is essential to formation of epithelial folds in the fly embryo. This is a model system of major importance in morphogenesis so their work is highly relevant for the field. The tools used are very well described and the quality of the data is really impressive. To date, all of the attention in ventral furrow formation has been on the apical surface, so these findings place new emphasis on events occurring basally. Overall, an outstanding effort worthy of publication in EMBO J. I recommend publication without delay.

Referee #2:

The paper by Krueger et al. examines the role of a reduction in basal myosin during the process of gastrulation in the *Drosophila* embryo. While extensive work was published on the organization and regulation of apical myosin, the untested assumption was that basal myosin should be decreased to allow normal gastrulation. The current paper utilizes state-of-the-art optogenetic approaches to increase the level of basal myosin at two different phases, and test this assumption. In the first phase they carry out targeted basal recruitment of RhoGEF2, and observe a non-autonomous effect on apical myosin constrictions. While the constrictions themselves continue to take place, they fail to be stabilized leading to loss of the ratcheting mechanism. The assumption is that the increased basal resistance to cell stretching affects the apical processes. In the second phase, local optogenetic activation was elegantly achieved by using a membrane protein that is basally located, rather than by local illumination. In this stage the excess basal myosin inhibited cell shortening and folding of the tissue to a tube-like structure.

The work is elegant and timely, and would be of interest to a wide audience. Publication is recommended after taking the minor comments below into consideration.

1. Establishing that the system works (Fig. 1): Not fully apparent what is being measured and why these measurements are important. In particular, panels 1J (z-spreading) and 1L (cell density). Authors should use simple, clear language in presenting these experiments.
2. Consequences of stabilizing basal myosin from the beginning of gastrulation (Figs 3 and 4): The manuscript could benefit from presenting this portion of the study in a somewhat different fashion. Rather than have the resulting "disorganized cell behavior" leading the way towards further analysis, move up the observation (Fig 4A) that the large majority of cells do not constrict their apices. Apical constriction is the cell-shape change most readily associated with ventral furrow formation, and the unexpected influence of stabilizing basal myosin on this process (along with the demonstration that this results from non-productive "ratcheting"), is perhaps the most novel observation stemming from the study.
3. Construction of a basally-targeted optogenetic probe (Fig 5): the authors construct a series of such probes, but although all localize basally, only one of them is "functional" (Fig 5E). No data accompanies the "functionality" observation. Do the other probes fail due to a lack of myosin recruitment? While expanding this section (perhaps as supplementary data) is not critical for the observations made and the conclusions drawn from using the active (CIBN-PatJ-CAAX) construct, it raises technical issues which will be of interest to future research using similar techniques.
4. It would be worthwhile to present data quantifying the degree of myosin recruitment following expression of CIBN-PatJ-CAAX in dorsal epithelium cells, similar to what was shown for the "generic" CIBN, to demonstrate a ~two-fold stabilization.

Referee #1:

This is an excellent manuscript. The authors use optogenetic activation of contractility to show that basal relaxation is essential to formation of epithelial folds in the fly embryo. This is a model system of major importance in morphogenesis so their work is highly relevant for the field. The tools used are very well described and the quality of the data is really impressive. To date, all of the attention in ventral furrow formation has been on the apical surface, so these findings place new emphasis on events occurring basally. Overall, an outstanding effort worthy of publication in EMBO J. I recommend publication without delay.

We thank this reviewer for the positive comments on our manuscript.

Referee #2:

The paper by Krueger et al. examines the role of a reduction in basal myosin during the process of gastrulation in the *Drosophila* embryo. While extensive work was published on the organization and regulation of apical myosin, the untested assumption was that basal myosin should be decreased to allow normal gastrulation. The current paper utilizes state-of-the-art optogenetic approaches to increase the level of basal myosin at two different phases, and test this assumption. In the first phase they carry out targeted basal recruitment of RhoGEF2, and observe a non-autonomous effect on apical myosin constrictions. While the constrictions themselves continue to take place, they fail to be stabilized leading to loss of the ratcheting mechanism. The assumption is that the increased basal resistance to cell stretching affects the apical processes. In the second phase, local optogenetic activation was elegantly achieved by using a membrane protein that is basally located, rather than by local illumination. In this stage the excess basal myosin inhibited cell shortening and folding of the tissue to a tube-like structure. The work is elegant and timely, and would be of interest to a wide audience. Publication is recommended after taking the minor comments below into consideration.

We thank this reviewer for the positive evaluation of our manuscript and useful suggestions

1. Establishing that the system works (Fig. 1): Not fully apparent what is being measured and why these measurements are important. In particular, panels 1J (z-spreading) and 1L (cell density). Authors should use simple, clear language in presenting these experiments.

In panel 1J z-spreading refers to the extent to which myosin-II diffuses away from the basal plane of photo-activation along the apico-basal axis. Because our aim was to achieve selective basal accumulation of myosin-II, this value represents an important indicator of the spatial precision at which we could control myosin-II levels. We have clarified the meaning of this measurement in the text (p. 8 lanes 8-12) and corresponding figure legend.

“During that time, myosin-II remained tightly localized at the basal surface of the cells, with a maximum spreading along the apico-basal axis (z-spreading) from the most basal plane of ~4 microns (Fig 1G-H and Movie EV3). This value was only

slightly higher than non-activated control cells (Fig 1J), demonstrating the efficacy of this protocol in the selective up-regulate of myosin-II at the base.

In panel 1L we replaced “cell density” with “ring density”. This value measures the density of actomyosin rings at the basal surface of the cells. We explained the meaning of this measurement in the corresponding figure legend and in the text (p. 8 last 7 lines) with the following paragraph: “To exclude the possibility that the increased actomyosin ring constriction caused a loss of basal tissue integrity, we compared the density of actomyosin rings inside and outside the photoactivation area. The result of this measurement revealed a ~1.3-fold increase in ring density upon photo-activation (Fig 1L and Movie EV2), arguing that cells were still interconnected at the base. Thus, we conclude that the established conditions allow for the quantitative control of myosin-II levels on the time-scale of ventral furrow formation”

2. Consequences of stabilizing basal myosin from the beginning of gastrulation (Figs 3 and 4): The manuscript could benefit from presenting this portion of the study in a somewhat different fashion. Rather than have the resulting "disorganized cell behavior" leading the way towards further analysis, move up the observation (Fig 4A) that the large majority of cells do not constrict their apices. Apical constriction is the cell-shape change most readily associated with ventral furrow formation, and the unexpected influence of stabilizing basal myosin on this process (along with the demonstration that this results from non-productive "ratcheting"), is perhaps the most novel observation stemming from the study.

While we agree that the inhibitory effects on apical constriction represents an important finding of our study, we would prefer to maintain the structure of our manuscript as it is. We have considered the possibility of changing the order in which Figs 3 and 4 are presented. However, we think that this change alters the flow of the manuscript with respect to the introduction and motivation of our study -which was to analyse the effects of basal myosin-II stabilization on cell shape changes in general and not specifically on apical constriction. We did not anticipate an inhibitory effect on ratchet constrictions but we came to this conclusion only after the quantitative analysis on cell shape presented in Fig.3. To highlight the inhibitory effect on apical constriction, we have included this finding both in the synopsis image and in the highlights of our manuscript.

3. Construction of a basally-targeted optogenetic probe (Fig 5): the authors construct a series of such probes, but although all localize basally, only one of them is "functional" (Fig 5E). No data accompanies the "functionality" observation. Do the other probes fail due to a lack of myosin recruitment? While expanding this section (perhaps as supplementary data) is not critical for the observations made and the conclusions drawn from using the active (CIBN-PatJ-CAAX) construct, it raises technical issues which will be of interest to future research using similar techniques.

We have added an entirely new Figure (Expanded Fig. 4) showing that -even upon one photon illumination, which provides a more powerful mean of photo-activation than two-photon- only the CIBN-PatJ-CAAX basal anchor induces myosin-II

recruitment and actomyosin ring constriction. The following paragraph has been added to the Result (p. 13, lanes 7-14) "In contrast, even upon one photon illumination during early cellularization stages, when the basal surface of the forming cells is only a few microns distant from the objective and thus more sensitive to photo-activation, the other basal anchors did not support myosin-II plasma membrane recruitment (Fig EV5A-L) nor actomyosin ring constriction during mid-cellularization (Fig EV5M-X). Thus, lack of myosin-II recruitment provides a mechanistic explanation of why only in the presence of CIBN-PatJ-GFP-CAAX could contractility be induced.

4. It would be worthwhile to present data quantifying the degree of myosin recruitment following expression of CIBN-PatJ-CAAX in dorsal epithelium cells, similar to what was shown for the "generic" CIBN, to demonstrate a ~two-fold stabilization.

To address this point, we have quantified myosin-II in ventral cells which demonstrates a two-fold increase, comparable to what achieved with the "generic" CIBN anchor both in dorsal and ventral cells. This data is presented in the new Expanded Fig. 4. and discussed in the Result (p. 13, lanes 3-7). "Consistent with this hypothesis, while CIBN-PatJ without a CAAX box did not support contractility, CIBN-PatJ plus the addition of a CAAX box, which upon prenylation is directly inserted into the lipid bilayer (Powers, 1991), resulted in a two-fold increase of myosin-II levels at the basal surface (Fig EV4) and increase in contractility (Fig 6A-C)."

2nd Editorial Decision

2nd Oct 2018

Thank you for submitting a revised version of your manuscript. It has now been seen by one of the original referees whose comments are shown below.

As you will see he/she finds that all criticisms have been sufficiently addressed and recommend the manuscript for publication. However, before I can officially accept the manuscript there are a few editorial issues concerning text and figures that I need you to address.

REFEREE REPORT:

Referee #2:

The authors have addressed our comments, and made compelling arguments in places where they preferred to stick to their original version. The paper is now ready for publication.

Corresponding Author Name: Stefano De Renzis

Journal Submitted to: The EMBO Journal

Manuscript Number: EMBOJ-2018-100170